# ColdGANs: Taming Language GANs
# with Cautious Sampling Strategies

**Thomas Scialom**$^{\star\ddagger}$**, Paul-Alexis Dray**$^{\star}$**, Sylvain Lamprier**$^{\ddagger}$**,**
**Benjamin Piwowarski**$^{\diamond\ddagger}$**, Jacopo Staiano**$^{\star}$
$^{\diamond}$ CNRS, France
$^{\ddagger}$ Sorbonne Université, CNRS, LIP6, F-75005 Paris, France
$^{\star}$ reciTAL, Paris, France
{thomas,paul-alexis,jacopo}@recital.ai
{sylvain.lamprier,benjamin.piwowarski}@lip6.fr

## Abstract

Training regimes based on Maximum Likelihood Estimation (MLE) suffer from known limitations, often leading to poorly generated text sequences. At the root of these limitations is the mismatch between training and inference, i.e. the so-called exposure bias, exacerbated by considering only the reference texts as correct, while in practice several alternative formulations could be as good. Generative Adversarial Networks (GANs) can mitigate those limitations but the discrete nature of text has hindered their application to language generation: the approaches proposed so far, based on Reinforcement Learning, have been shown to underperform MLE. Departing from previous works, we analyze the exploration step in GANs applied to text generation, and show how classical sampling results in unstable training. We propose to consider alternative exploration strategies in a GAN framework that we name $ColdGANs$, where we force the sampling to be close to the distribution modes to get smoother learning dynamics. For the first time, to the best of our knowledge, the proposed language GANs compare favorably to MLE, and obtain improvements over the state-of-the-art on three generative tasks, namely unconditional text generation, question generation, and abstractive summarization.

## 1 Introduction

Deep learning approaches have paved the way for significant achievements in Natural Language Generation (NLG). Under the most popular paradigm, sequence to sequence models [40] are trained with Maximum Likelihood Estimation (MLE) via Teacher Forcing [50]. Training neural networks under MLE does not succeed in modeling sequence probabilities [48], since, at inference, the model is conditioned on sequences that may have never been observed at training time. Indeed, generated texts using this approach are often *degenerate* [16], e.g. prone to repetition.

Nonetheless, these same architectures, when used as discriminators, are able to distinguish human from machine-generated text with a disconcerting efficiency: reported values are around 97% for long article generation [53] or abstractive summarization [37]. In the generative architectures, the encoder part can reach such performances, supporting the hypothesis that generation failures are mostly due to the decoding step: under MLE training regimes, the decoding suffers from exposure bias [33, 1] and lacks a sequence-level loss to optimize [26].

To mitigate MLE limitations, Reinforcement Learning (RL) has been applied to text generation tasks [33, 29], considering sequence level metrics such as BLEU or ROUGE as the reward. However, such metrics, based on $n$-grams similarity, are known to poorly correlate with human judgments [27], and do not preserve meaning [39]. Hence, when reinforced on them, models yield to poorer genera-

tions and higher degradation compared to their MLE counterparts [29]. To overcome these drawbacks, better rewards are thus necessary [29].

To this end, Ziegler et al. [60] proposed to directly reward systems using human judgment. Although this approach performs very well and approximates the best possible reward, it is obviously not a viable solution in practice. However, it attests that, with perfect rewards, one can achieve excellent levels of performance. A natural alternative, not requiring human judgments, is to frame the problem under the Generative Adversarial Network (GAN) paradigm [13], which has been used successfully for image generation [2]. For text, modeled as a sequence of discrete symbols, a naive computation of the gradients is however intractable. Hence, Language GANs are based on gradient estimation via RL-based techniques [52].

However, the reward in this case can be extremely sparse (as discussed in Section 3.2), yielding to high-variance gradient estimation, which is known to be challenging for optimization [56]. Most previous works have focused on this aspect, and proposed denser rewards [19, 22]. Unfortunately, these attempts to apply GANs to text generation obtained limited success [4] and have been found to underperform MLE [38, 42, 22].

Although known to be crucial [41], *exploration* is surprisingly understudied when RL is applied to text generation. In this work, we propose a new exploration method that aims at sampling more structured rewards and that better suits the GANs' training dynamics, allowing for the first time to successfully train Language GANs. Our main contributions can be summarized as:

1. We study the discriminators' behavior and show that their degree of specialization has important implications on the exploration to stabilize the training process. In particular, we find that reducing the exploration space is essential to successfully train discrete GANs.

2. Based on these observations, we propose $ColdGANs$, a GAN architecture using alternative sampling strategies that force the sampling to remain closer to the distribution modes.

3. Finally, we apply our proposed methods on three tasks. We report positive results compared to previous works, including GANs and MLE-based models.

## 2 Related Work

**RL for text generation**    Since many metrics of interest in NLP are non-differentiable, several approaches used RL for text generation [7, 33, 29, 5]. To our knowledge, all works based on RL for text generation use standard sampling for policy gradient estimation, following the current policy from the generator they define. Apart from text GANs, they all suffer from the aforementioned limitations of ill-defined reward metrics, such as BLEU or ROUGE [29].

**Text GANs**    Tackling this problem by implicitly learning the metric via a discriminator, adversarial approaches have been proposed for text generation. Given the very high dimension of the generative (action) space, and the sparsity of associated rewards provided by the discriminator (see Section 3.2), a large body of works focused on defining denser rewards: ranking and comparative discriminators [19, 58], sequential discriminators where the rewards are provided at each time step of the generation [38, 22], or using masked language modeling [10]. The policy is usually learned via vanilla Policy Gradient REINFORCE [49], with the exception of MaliGAN [6], which deals with the problem of the discriminator being a moving target trough Importance Sampling (IS). Another difficulty with GANs for discrete sequential data is that discriminators are inaccurate for samples close to the generator distribution modes, as those used for training are usually too scattered over the full space to enable specialization on useful/difficult areas (see Section 3.2 for preliminary experiments on this).

**Cautious RL**    Standard works in RL proposed ways to avoid catastrophic moves of the policy parameters [35, 36], by enforcing the new policy to stay close to the current one at each step via KL-divergence constraints. In this work, our main focus is rather to stay close to the *comfort zone* of the reward function, which becomes easily noisy outside, independently of the policy.

**Importance Sampling for Reinforcement Learning**    In RL, IS is generally used for sample efficiency purposes: in off-policy policy gradient methods, IS allows to re-use previously sampled sequences more than once [47, 43, 11]. Conversely, in this work, IS is employed to improve the

stability of RL for Text GANs. Closer to our work, MaliGAN [6] proposes to rely on IS to consider an estimation of the data distribution as a target (via a KL objective). Although theoretically appealing, its stability relies on very strong assumptions about discriminator guarantees, which rarely hold in practice. Instead, we propose to rely on IS to stabilize the generator-discriminator min-max game via alternative careful sampling strategies. Note also that our approach could easily be included in the MaliGAN framework.

## 3 Discriminators and Generators Interaction

### 3.1 Generating and discriminating as text to text tasks

**Generator** Text generation naturally lends itself to autoregressive modeling [40]. The probability to generate a sequence $Y$ composed of $N$ tokens $y_1, ..., y_N$ is given by:

$$p_\theta(Y|X) = \prod_{t=1}^{N} p(y_t|y_1, ..., y_{t-1}, X, \theta) \tag{1}$$

where $\theta$ are the learnable parameters of the generator and $X$ the input sequence.

Neural networks typically produce class probabilities by using a "softmax" output layer that converts the logit $z_i$, computed for each token of the vocabulary, into a probability $q_i$:

$$q_i = \frac{\exp(z_i/T)}{\sum_j \exp(z_j/T)} \tag{2}$$

where $T$ is a "temperature" hyper-parameter, set to 1 unless otherwise specified. The higher the temperature, the more uniform the probability distribution over the vocabulary, resulting in more diversity but also more mistakes [15]. In the following, we note as $\pi_\theta$ the distribution defined by the generator with temperature $T = 1$.

**Discriminator** In the following, we consider a discriminator $D_\phi$ learned from sets of human and generated texts for each input $X$ as a logistic regression problem:

$$\frac{1}{|H|} \sum_{(X,Y) \in H} \log(D_\phi(X,Y)) + \frac{1}{|G|} \sum_{(X,Y) \in G} \log(1 - D_\phi(X,Y))$$

where $H$ is a set of pairs of input $X$ associated with a human written text $Y$ from the data distribution, and $G$ is a set of pairs with generated outputs $Y$.

**Text to text tasks** Casting any NLP task as a text-to-text problem, T5 [31] demonstrated state-of-the-art results on the established GLUE benchmark [46] and on its more challenging successor [45]. Accordingly, we employ the same architecture for both discrimination and generation. This allows for fairer comparisons thereafter, as both generator and discriminator have the same architecture, pre-training and capacity.

### 3.2 Discriminator-Generator Equilibrium

**Exposure Bias** As mentioned above, a discriminator can easily predict the human or machine nature of a text. One reason for this lies in exposure bias. To quantify this statement, we compare the results for a discriminator when trained under the two following generation strategies: *Standard Generation*, suffering from the exposure bias; and, *Teacher Forcing Generation*, where the ground-truth tokens $y_{i<t}$ are fed to the generator, so not to expose the model to its own prediction, and only $y_t$ is generated by a machine.

We show the results in Fig. 1. As expected, the two discriminators have the same score for $t = 0$. We observe that both perform well, and that the Standard Generation discriminator obtains consistently larger improvements, w.r.t. the Teacher Forcing Generation discriminator, as the length of the sequence increases. This could indicate the presence of the exposure bias, for which the errors accumulate over time. Still, the relatively high accuracy observed under Teacher Forcing Generation suggests that additional factors, beyond exposure bias, might be involved: in the following, we show that the extreme specialization of discriminators is among those.

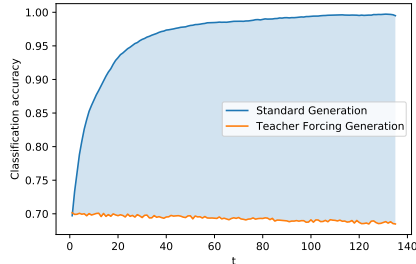

Figure 1: Accuracy of a discriminator model trained under two different generation modes: *Standard* (subject to the exposure bias) and *Teacher Forcing*. The x-axis corresponds to the partial length $t$ of the sequence to discriminate.

Table 1: Probability that a text is human according to various discriminators. $D_{perfect}$ corresponds to a theoretical perfect discriminator with infinite capacity and training data. $D_{T=\gamma}$ corresponds to a discriminator trained on samples generated with a temperature $T = \gamma$. Past $T = 0$ and past $T = 1$ correspond to results on samples obtained with the generator weights resumed from a previous stage of the training, i.e. a checkpoint one epoch before the final state (see Section 4, Memory Replay).

|  | Evaluated on | | | | | |
| --- | --- | --- | --- | --- | --- | --- |
|  | human | $T = 0$ | $T = 1$ | $T = \infty$ | past $T = 0$ | past $T = 1$ |
| $D_{T=0}$ | .79 | .17 | .84 | .92 | .26 | .85 |
| $D_{T=1}$ | .79 | .76 | .23 | .09 | .75 | .31 |
| $D_{T=\infty}$ | .92 | .92 | .91 | .08 | .92 | .91 |
| $D_{T\in\{0,1,\infty\}}$ | .69 | .24 | .24 | .09 | .32 | .36 |
| $D_{perfect}$ | 1 | 0 | 0 | 0 | 0 | 0 |

**Discriminator's No Free Lunch** As defined above, the temperature $T$ of the generator is a hyper-parameter which allows to control the randomness of predictions while sampling, by scaling the logits before applying a `softmax`. Thus, we can define various sampling strategies from the same generator. Low (close to 0) temperatures provide samples close to the sequence $s_\theta^{greedy}$ of a greedy procedure that takes the token with max generator probability $\pi_\theta$ at each step (the output of a beam search with beam size $B = 1$). With high temperatures, the distribution of sequences tends to the uniform distribution. We experiment with different temperature settings for the same generator (trained with MLE), and use the obtained samples to train and test a discriminator. This allows us to evaluate the impact of differences in sampling temperatures, between training and inference, on the discriminator performance. In other words, how a discriminator, trained with samples obtained at a specific temperature, performs when faced with samples generated under different sampling setups.

We train and evaluate discriminators on samples generated under temperatures $T = 0$, $1$ or $\infty$, for a conditional generation task (summarization, see Section 5.2), which allows to consider various sequence samples even at low temperatures. We report the results in Table 1. As expected, in all but one case, discriminators perform better if trained and evaluated with sequences generated under the same temperature (no mismatch). However, when the training and evaluation samples are generated with different temperatures, we observe that the discriminator fails to distinguish human from generated ones. More precisely, it considers most sentences to be human-generated (around 90%). Conversely, when trained on the different temperatures together ($T \in \{0, 1, \infty\}$), results are more balanced: robust across the various temperatures, but yielding a drop in accuracy, consistently with the well-known accuracy-robustness trade-off [12, 3]. This highlights that individual discriminators are specialized on specific generation pairs (machine/human). Knowing this, it is crucial to orient this specialization on useful areas.

Interestingly, when trained from samples issued from $\pi_\theta$, the discriminator $D_{T=1}$ is inaccurate at identifying samples close to $s_\theta^{greedy}$ as generated ones: $D_{T=1}(s)$ equals 0.76 on average over these samples. This is particularly bad for a discriminator used as a reward signal of a RL process, since such samples lie in the useful area of the output distribution. They correspond to samples close to the modes of the distribution $\pi_\theta$. Moreover, in many text generation applications, generation

strategies such as beam search target these sequences as prediction outputs. A bad reward function at these locations is likely to lead to bad generation performance. Besides, the discriminator trained on samples close to the mode of $\pi_\theta$ (i.e., $D_{T=0}$) is bad for samples from $\pi_\theta$ (i.e., $T = 1$), indicating that one cannot simply use such samples to train the discriminator while considering standard sampling for generator training (as rewards would be very inaccurate).

**Implications for Discrete GANs**    Holtzman et al. [16] report that for $T = 1$, sampling from the tail of the distribution is expected to happen within the first three steps of decoding and with a probability superior to 99.96% within 20 steps. Such unstructured exploration causes a large variance which grows with the number of time steps, and perturbs actions too frequently [34, 17]. A less random exploration would thus yield to better structured sequences and lower variance, closer to the distribution learned by the discriminator, and would likely enable better training dynamics between the discriminator and the generator.

## 4   Models

Based on the findings above, we seek sampling strategies that allow both the discriminator to train on useful samples, and the generator to be trained from reliable rewards from the discriminator, within a policy gradient RL scheme where we are interested at maximizing $J(\theta) = \mathbb{E}_{\tau \sim \pi_\theta}[D_\phi(\tau)]$, according to generator parameters $\theta$. The discriminator is updated at the end of each training epoch, via gradient ascent on human-machine pairs, with new artificial sequences resulting from the generator distribution. In order to introduce cautious sampling that focuses more on modes of distributions, note that it would be useless to consider the policy gradient $\nabla_\theta \mathbb{E}_{\tau \sim \pi_\theta^{T=\gamma}}[D_\phi(\tau)] = \mathbb{E}_{\tau \sim \pi_\theta^{T=\gamma}}[D_\phi(\tau)\nabla_\theta \log \pi_\theta^{T=\gamma}(\tau)]$ of a generator distribution with modified temperature $T = \gamma$, as it would, compared to $T = 1$, only imply rescaling the network outputs without altering the learning process.

Instead, we propose to employ Importance Sampling for defining our cautious sampling strategies for text GANs, based on the fact that, for any distribution $P, Q : \mathcal{X} \to [0, 1]$ such that $Q(x) > 0$ whenever $P(x) > 0$, and any function $f : \mathcal{X} \to \mathbb{R}$, we have $\mathbb{E}_{x \sim P(x)}[f(x)] = \mathbb{E}_{x \sim Q(x)}[\frac{P(x)}{Q(x)}f(x)]$. In our case, this yields the following unbiased policy gradient:

$$\nabla_\theta J(\theta) = \mathbb{E}_{\tau \sim \hat{\pi}_\theta} \left[ \frac{\pi_\theta(\tau)}{\hat{\pi}_\theta(\tau)} D_\phi(\tau) \sum_{t=1}^{|\tau|-1} \nabla_\theta \log \pi_\theta\left(\tau_t | \tau_{1:t-1}\right) \right] \qquad (3)$$

where $\tau_t \in V$ is the $t$-th token from sequence $\tau$ and $\tau_{1:t-1}$ the subsequence of its $t - 1$ first tokens, $\pi_\theta$ the generator probability and $\hat{\pi}_\theta$ a modified sampling distribution, which enables the generation of any possible sequence of tokens given the vocabulary $V$.

In this work, we focus on the exploration stage; therefore, conversely to previous works, we can choose the most sober form of reward: 1 if $D_\phi(\tau)$ predicted human, and 0 otherwise. We show that a sparse reward is not a limitation if the sampling strategy is close to the modes of the distribution – provided the initial solution is a good enough bootstrap (which, according to our experiments, is the case). Note that $D_\phi$ is trained with samples from $\hat{\pi}_\theta$ to avoid any mismatch with the generator training samples, which would be problematic otherwise (as pointed out in Section 3.2).

**ColdGANs exploration**    The temperature $T$ plays a major role in moderating exploration. Indeed, being a scaling factor applied to the generator outputs, it directly defines the degree of diversity of the generated sequences. The default exploration is obtained by recursively sampling a sequence of tokens from the model distribution with $T = 1$. The higher $T$, the more random the sampled sequences, regardless of the model's policy. Conversely, lower temperatures reduce the exploration, with $T \to 0$ ultimately equivalent to the `argmax` function. Therefore, we consider a distribution $\hat{\pi}_\theta = \pi_\theta^T$ with lower (*colder*) temperatures $T \in ]0, 1[$. This allows to explore sequences composed of tokens less likely to be sampled from $\hat{\pi}_\theta$ tail. Note that for $T > 0$, $\hat{\pi}_\theta > 0$ whenever $\pi_\theta > 0$.

**ColdGANs$_{nucleus}$**    In addition, we consider a more sophisticated technique: *nucleus sampling* [16]. This decoding method has been shown to produce higher quality texts than previous sampling strategies, including those temperature-based. Sampling from the nucleus of tokens containing the

vast majority of the probability mass, the approach dynamically truncates the unreliable tail of the probability distribution and hence is an instance of a *cautious* generative process. However, with nucleus sampling, many sequences $\tau$ get $\hat{\pi}_\theta(\tau) = 0$ while $\pi_\theta(\tau) > 0$, invalidating the IS. To avoid this, we propose to use a mixture combining low temperatures and nucleus policies:

$$\hat{\pi}_\theta(\tau) = \epsilon\pi_\theta^{nucleus}(\tau) + (1 - \epsilon)\pi_\theta^{T=\gamma}(\tau) \qquad (4)$$

where $\epsilon$ is a hyper-parameter, $\pi_\theta^{nucleus}$ is the probability under nucleus and $\pi_\theta^{T=\gamma}$ the probability rescaled for temperature $\gamma$, as described in the previous paragraph.

**Importance Weight Clipping**  The importance weights can become large, causing instability. Adapting from [47] (see paragraph 3.2 of their paper for more details), we truncate the importance weights and add a correction term in the computation of $\nabla_\theta J(\theta)$:

$$\mathbb{E}_{\tau \sim \hat{\pi}_\theta}\left[\min(c, w(\tau))D_\phi(\tau)\nabla\log\pi_\theta(\tau)\right] + \mathbb{E}_{\tau \sim \pi_\theta}\left[\max\left(0, \frac{w(\tau) - c}{w(\tau)}\right)D_\phi(\tau)\nabla\log\pi_\theta(\tau)\right]$$

where $w(\tau) = \frac{\pi_\theta(\tau)}{\hat{\pi}_\theta(\tau)}$. In the first term of Eq. 4, by clipping the importance weight, the variance of the gradient estimate is bounded. The second term of the equation ensures that our estimate is unbiased, by re-sampling another sequence from the true policy $\pi_\theta$. In our experiments, we set $c = 5$. Note that, contrary to off-policy RL, for which such a IS clipping was proposed [47], in our case clipping is very rare: it only occurs for sequences whose probability from the generator is much higher than the one from the sampling distribution, which is designed for sampling close to the mode of $\pi_\theta$. However, if this happens, this clipping ensures that the corresponding gradient does not explode.

**Memory Replay**  In Table 1, we observed that the performance of the discriminators is lower when evaluated on samples generated from the previous checkpoint of the same model (i.e., evaluated on past $T$). We connect this to the failure mode in GANs observed by Metz et al. [24], where the generator and the discriminator oscillate during training, rather than converging to a fixed point. In lifelong learning literature [23], it has been shown that 1% of experience replay is sufficient to avoid catastrophic forgetting. Inspired by this work, we construct a memory buffer which contains samples generated in the last $K$ training steps, and replace 1% of the discriminator training examples with samples from the buffer. This allows the discriminator to remain accurate on the samples from the previous state of the generator, hence preventing such failure loop during training.

## 5 Experiments

Due to the computational cost of T5-large (11B parameters), we used T5-small (60M parameters). For all our experiments, we used the validation sets for hyperparameter selection. In more detail, we evaluated our approach with several learning rates,[1] reporting results for a value of 2e-5. From the best performing $ColdGAN$ configuration, we perform ablations to assess the impact of Memory Replay and Importance Weight Clipping. Finally, we experimented with BART [18] instead of T5.[2]

### 5.1 Unconditional Language Generation

Most previous works for language GANs have been evaluated on unconditional language generation benchmarks. In this task, no input is provided and the goal is to generate both meaningful and diverse texts. Consistently with [22], we measure these two aspects using, respectively, BLEU [28] and self-BLEU [59] metrics.[3] the To obtain a finer comparison between models, Caccia et al. [4] proposed to draw the curve of (negative) BLEU vs self-BLEU, by sampling with various temperatures at inference. This allows to measure the trade-off between quality and diversity. Following [6, 20, 38, 14, 4, 22], we used the EMNLP2017 news dataset.[4] We report $ColdGANs$ results in Figure 2 (left). Notice that previous works did not use self-supervised pretrained models, while we did (with T5): this explains

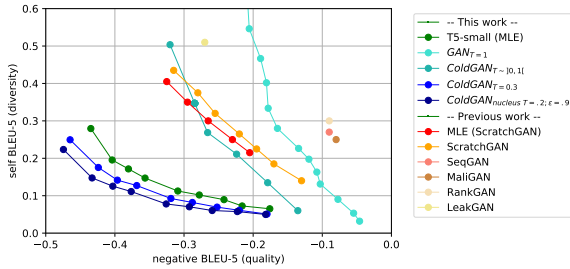

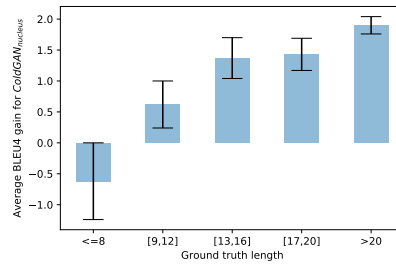

Figure 2: Results on the EMNLP 2017 News dataset (for all metrics, lower is better). Scores for previous works are taken from [22].

Figure 3: Relative BLEU-4 gains obtained with $ColdGANs$ over MLE, grouped by ground truth sequence length, on QG.

the improvement of our MLE baseline over theirs (MLE ScratchGAN). As one cannot directly compare our performances with those reported from previous works, we study the performance variations from the corresponding MLE baseline. Consistently with previous works [38, 42, 22], we observe that the model, under the default exploration (i.e. $GAN_{T=1}$), performs strictly worse than MLE. As a baseline, we experimented $ColdGAN_{T\sim]0,1[}$, where during the training the temperature is randomly sampled between 0 and 1 for each sequence. While it performs better than $GAN_{T=1}$, it still does not compare favorably w.r.t. MLE. We report the results for a T5 model trained with the ScratchGAN protocol, and found it did not compare favorably w.r.t. T5 (MLE). Conversely, both $ColdGAN_{T=0.3}$ and $ColdGAN_{nucleus}$ obtain better results than MLE for the entire curve. To our knowledge, this is the first time that *MLE falls short* [4, 22] w.r.t. GAN-based approaches for this task.

## 5.2 Conditional Language Generation

We evaluate $ColdGANs$ on two popular tasks where text inputs are given for conditioning the generation, namely Question Generation and Text Summarization. These are highly competitive benchmarks, with recent state-of-the-art results achieved by MLE based on pre-trained transformers [44]. Answer-aware Question Generation (QG) [57] is the task wherein, given a text and a target answer, the goal is to generate a relevant question. Following previous works [8, 9], we used the SQuAD dataset [32]. Automatic Summarization aims to produce concise and fluent summaries given a longer text. We used the popular CNN/DM dataset [25], a corpus containing news articles and the corresponding abstractive summaries. For conditional text generation tasks, output sequences are commonly evaluated using BLEU (for e.g. Machine Translation, Question Generation) or ROUGE (for e.g. Summarization) metrics. In contrast to the unconditioned scenario, the diversity is linked to the variety of the inputs, and it is common practice to decode through beam search at inference.

**Results** For both tasks, we used data and evaluation metrics released by Dong et al. [8].[5] The results shown in Table 2 are consistent across the two tasks: again, we observe that exploring under the default temperature yields to poor performances, while $ColdGANs$ compare favorably to MLE. The best performance is achieved with the experiment emphasizing the $ColdGAN_{nucleus}$ exploration the most, with $\epsilon = .9$ and $T = .2$. Over 10 independent training runs, we also observed very stable results for this model, with a standard deviation of the average BLEU-4 lower than .09 on the test set. Finally, we applied this last $ColdGANs$ setup to BART [18], achieving a new state-of-the-art on both QG with 23.05 BLEU-4 and summarization with 41.12 ROUGE-L.

**Mitigating the Exposure Bias** In Figure 3 we report the relative gain obtained, in terms of BLEU-4 for T5-small, for the best configuration (i.e. $ColdGAN_{nucleus}$, $\epsilon = 0.9$) w.r.t. the corresponding MLE baseline. The x-axis gives the length of considered ground truth target sequences. We observe that the longer the target sequence, the more the $ColdGAN$ outperforms MLE. This might indicate that $ColdGANs$ can successfully mitigate exposure bias.

Table 2: Results on Question Generation (QG) and Abstractive Summarization (Summ.) tasks.

| | #params | QG (SQuAD) | | Summ. (CNN/DM) | | |
| | | BLEU-1 | BLEU-4 | ROUGE-1 | ROUGE-L | BLEU-4 |
| --- | --- | --- | --- | --- | --- | --- |
| SemQG [55] | | | 18.37 | | | |
| BertSumAbs [21] | 340M | | | 41.72 | 38.76 | |
| UniLM [8] | 340M | | 22.78 | 43.33 | 40.41 | |
| PEGASUS [54] | 568M | | | 44.17 | **41.11** | |
| T5-large (MLE) [31] | 11B | | | 43.52 | 40.69 | |
| T5-small (MLE) [31] | 60M | 47.72 | 19.65 | 42.34 | 40.37 | 15.94 |
| " $(GAN_{T=1})$ | 60M | 46.44 | 18.84 | 38.98 | 36.42 | 13.23 |
| " $(ColdGAN_{T=.2})$ | 60M | 47.94 | 20.23 | 42.58 | 40.74 | 16.04 |
| " $(ColdGAN_{nucleus\ T=1;\epsilon=.1})$ | 60M | 46.82 | 18.97 | 39.05 | 38.01 | 14.04 |
| " $(ColdGAN_{nucleus\ T=1;\epsilon=.9})$ | 60M | 47.83 | 20.85 | 42.31 | 40.44 | 16.21 |
| " $(ColdGAN_{nucleus\ T=.2;\epsilon=.9})$ | 60M | 48.50 | 20.55 | 42.54 | 40.61 | 16.86 |
| *w/o Memory Replay* | 60M | 48.93 | 20.52 | 42.34 | 40.44 | 16.72 |
| *w/o IS Weight Clipping* | 60M | 48.21 | 20.14 | 42.23 | 40.35 | 16.72 |
| BART (MLE) [18] | 400M | 53.13 | 22.68 | 44.16 | 40.90 | 17,87 |
| " $(ColdGAN_{nucleus\ T=.2;\epsilon=.9})$ | 400M | **53.73** | **23.05** | **44.46** | **41.12** | **18.17** |

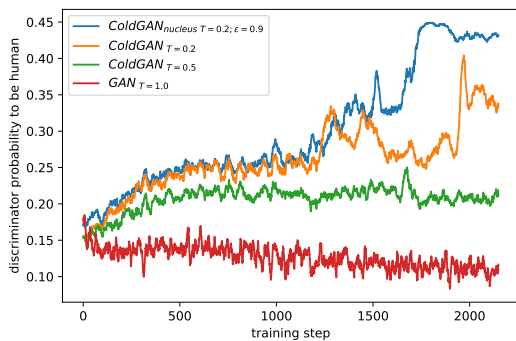

Figure 4: Probability that the generated text is human according to $D_\phi$ on CNN/DM.

Table 3: Human evaluation on QG. $ColdGAN$ corresponds to BART trained with $ColdGAN_{nucleus}$ $T = .2; \epsilon = .9$. Two-tailed t-test results are reported for each model compared to Human (*: $p < .01$, **: $p < .001$).

| | Fluency | Relevance | Answerability |
| --- | --- | --- | --- |
| Human | 3.66 | 4.31 | 4.22 |
| BART (MLE) | 3.80* | 4.43 | 4.11 |
| $ColdGAN$ | **4.36**** | 4.45 | 4.01 |

**Human Evaluation** As discussed in Section 1, automatic metrics are known to suffer from key limitations. Therefore, we additionally conducted a human evaluation on the QG task. Three professional English speakers were asked to judge, on a 1-to-5 Likert scale, to what extent the generated questions were: well-posed and natural (*Fluency*), relevant to their context (*Relevance*), and answerable, by looking at their context and answer (*Answerability*). The results in Table 3 show, surprisingly, both MLE-BART and $ColdGAN$-BART outperform the ground truth for Fluency. A similar result was reported by Yoon et al. [51] (refer to Table 2 in their paper). A plausible explanation is that humans are more inclined to use informal language and make grammar mistakes. For instance the human question *"About how many yellow cabs operate in New York?"* sounds slightly less formal than the one, generated by $ColdGAN$, *"How many yellow taxicabs are in Manhattan ?"*. Compared to MLE, $ColdGAN$ enables to significantly improve in term of fluency, while remaining competitive on other metrics, consistently with our experiments on exposure bias.

**Adversarial training curves** Figure 4 shows the evolution (during training and for different setups) of the probability of the generated text to be human, according to the discriminator. Consistently with Table 2, $ColdGAN_{nucleus}$ appears to be the most adverse to the discriminator. Conversely, the regular GAN ($T = 1$) is less and less adversarial, and comparatively more perturbed.

# 6   Conclusion

We proposed $ColdGANs$, a novel approach able to *tame* the exploration in Language GANs, allowing to obtain performance improvements on both conditional and unconditional text generation, w.r.t to MLE-based training. Our proposed IS method makes it compatible with advanced sampling methods,

such as nucleus, or other future decoding methods. In the future, we plan to combine $ColdGANs$ with orthogonal approaches proposed by previous works, such as denser rewards.

## Broader Impact

Fluent and reliable Natural Language Generation can have significant societal impacts. On the one hand, we envision several applications beneficial for business, research or education: from automatic summarization of news, papers or books, to efficient information access; from automatic and personalized student evaluation tests trough question generation, to responsive conversational interfaces. On the other hand, malicious actors can use the same technology to build tools detrimental to society, e.g. for creation and propagation of misleading (fake) news as discussed in [30], impersonation, and deceit. Nonetheless, keeping this research open and under public scrutiny is arguably one of the best ways to defend against such actors [53].

## Funding Transparency Statement

**Funding (financial activities supporting the submitted work)** : Funding in direct support of this work by the company ReciTAL, and the scholarship from Association Nationale Recherche Technologie (ANRT) for Thomas Scialom's thesis.

**Competing Interests (financial activities outside the submitted work)** None of the authors perceived additional revenues related to this work.

## Footnotes

[1]2e-6, 8e-6, 2e-5, 8e-5, 2e-4.

[2]BART has comparable performance to T5-large, but with 20x fewer parameters.

[3]Implemented in `https://github.com/deepmind/deepmind-research/tree/master/scratchgan`

[4]`http://www.statmt.org/wmt17/`

[5]https://github.com/microsoft/unilm/tree/master/unilm-v1

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
