[Supplementary Material]

# ColdGANs: Taming Language GANs with Cautious Sampling Strategies Supplementary Material

**Thomas Scialom**[⋆‡], **Paul-Alexis Dray**[⋆], **Sylvain Lamprier**[‡],
**Benjamin Piwowarski**[◇‡], **Jacopo Staiano**[⋆]
[◇] CNRS, France
[‡] Sorbonne Université, CNRS, LIP6, F-75005 Paris, France
[⋆] reciTAL, Paris, France
{thomas,paul-alexis,jacopo}@recital.ai
{sylvain.lamprier,benjamin.piwowarski}@lip6.fr

## 1  Implementation Details

All models are implemented in PyText [1]. We used a single RTX 2080 Ti GPU. All our experiments were conducted with T5-small [1] (60 million parameters) for both the generator and the discriminator; these were first trained on the corresponding task with MLE as in [2]. While T5-small underperforms its larger version, T5-11B, the latter has 11 billion parameters. However, BART [3] performs as well with only 400M parameters. Hence, for each task, we chose to train BART following the same procedure, with the best set of hyper-parameters found with T5-small (i.e. $ColdGAN_{nucleus\ T=.2;\epsilon=.9}$). For T5 and BART in conditional text generation, we applied at inference beam search with K=1 for T5 and K=4 for BART as recommended [3]. For Question Generation, One epoch to train ColdGAN takes 2 hours with T5 and 5 hours with BART, twice more for Summarization and less than a hour for Unconditional Text Generation.

## 2 Output Examples

In the samples below, **Human** corresponds to the ground truth; ***MLE*** to the baseline trained with Teacher Forcing; ***ColdGAN***, the model for which we obtain the best results (i.e. $ColdGAN_{nucleus\ T=.2;\epsilon=.9}$)

**Question Generation examples on SQuAD**

.

---

**CONTEXT:** A HDI below 0 . 5 is considered to represent " low development " . All 22 countries in that category are located in Africa . The highest - scoring Sub - Saharan countries , Gabon and South Africa , are ranked 119th and 121st , respectively . Nine countries departed from this category this year and joined the " medium development " group .
**ANSWER:** 119th

**HUMAN:** What is Gabon ' s ranking ?

*ColdGAN:* What is Gabon ' s rank on the HDI ?

*MLE:* Where is Gabon ranked in the Sub - Saharan category ?

---

**CONTEXT:** In 2012 , New York City had the lowest overall crime rate and the second lowest murder rate among the largest U . S . cities , having become significantly safer after a spike in crime in the 1970s through 1990s . Violent crime in New York City decreased more than 75 % from 1993 to 2005 , and continued decreasing during periods when the nation as a whole saw increases . By 2002 , New York City ' s crime rate was similar to that of Provo , Utah , and was ranked 197th in crime among the 216 U . S . cities with populations greater than 100 , 000 . In 2005 the homicide rate was at its lowest level since 1966 , and in 2007 the city recorded fewer than 500 homicides for the first time ever since crime statistics were first published in 1963 . In the first six months of 2010 , 95 . 1 % of all murder victims and 95 . 9 % of all shooting victims in New York City were black or Hispanic ; additionally , 90 . 2 percent of those arrested for murder and 96 . 7 percent of those arrested for shooting someone were black or Hispanic . New York experienced a record low of 328 homicides in 2014 and has a far lower murder rate than other major American cities .
**ANSWER:** 2007

**HUMAN:** In what year did the city have less than 500 homicides ?

*ColdGAN:* In what year did New York City record fewer than 500 homicides for the first time since 1963 ?

*MLE:* When did New York City record fewer than 500 homicides ?

---

**Summarization examples on CNN/DM**

**CONTEXT:** A couple of wartime sweethearts who were married for almost 72 years have been buried next to each other after dying days apart. Ronald Pearson and his wife Miriam met at an evening dance while he was serving in the RAF police and she the Auxiliary Territorial Service during the Second World War. They married in 1943 and settled in Broughton near Chester, welcoming their daughter two years after celebrating VE day. After almost 72 years together, Mrs Pearson, 95, died last month. Her husband followed two days later aged 94. Ronald and Miriam Pearson died last month within two days of each other. The couple, who met during the war, had been married for almost 72 years . Mr and Mrs Pearson met during the War when he was working as a sergeant for the RAF police and she a driver for the Auxiliary Territorial Service . Their marriage was described as 'the greatest true love story' at a joint funeral held at Blacon Crematorium which saw the couple buried side by side. 'It's nice that they are going together; it's what they would have wanted. They'll be next to each other, like they always were,' said their daughter, Jenny Ledger, 68.

**HUMAN:** Ronald Pearson and his wife Miriam met and were married during WWII . He was a sergeant in the RAF police while she worked as a driver for ATS . The 'inseparable' couple settled in Broughton near Chester to raise family . They died last month within two days of each other at the ages of 94 and 95 . Their marriage was described as 'greatest true love story' at joint funeral .

*ColdGAN:* Ronald and Miriam Pearson met during the War when he was serving in the RAF police and she the Auxiliary Territorial Service . They married in 1943 and settled in Broughton near Chester , welcoming their daughter two years after celebrating VE day . After almost 72 years together , Mrs Pearson , 95 , died last month . Her husband followed two days later aged 94 .

*MLE:* Ronald and Miriam Pearson met during the War when he was a sergeant . They met at an evening dance while he was serving in the RAF police . They married in 1943 and settled in Broughton near Chester . Mrs Pearson , 95 , died last month after 72 years together . Her husband followed two days later aged 94 .

---

**CONTEXT:** He is a World Cup winner, Spanish football legend, and one of the most recognisable faces in the classy Barcelona CF outfit. And now tourists are being offered the chance to walk in the footsteps of Andrés Iniesta after he listed his beautiful Spanish vineyard on Airbnb. The world class midfielder took to Twitter to advise that his the 'Bodega Iniesta' vineyard he owns in Castilla-La Mancha can be rented out. Spain and Barcelona midfielder Andrés Iniesta is renting out his vineyard on Airbnb . Andrés Iniesta posted on Twitter to spread the news that his charming vineyard can be rented out . And it's a snip of a price to tread paths made by one of the beautiful game's best players at €125 (£90) a night. There is one bathroom and one bedroom at the charming little property, with other facilities including a kitchen, an indoor fireplace, Wi-Fi and parking provided. The residence is aimed for couples as the bedroom consists of one double bed.

**HUMAN:** Spanish football legend took to Twitter to announce the listing . Bodega Iniesta' vineyard is located in Castilla-La Mancha, Spain . Property has one bedroom with double bed, and kitchen and lounge area . Guests are warned not to be too physical with the vines .

*ColdGAN:* Andrés Iniesta listed his vineyard on Airbnb . The Spanish midfielder posted on Twitter to spread the news . The vineyard is a snip of a price to tread paths made by one of the beautiful game 's best players at $ 125 (£ 90) a night .

*MLE:* Spain and Barcelona midfielder Andrés Iniesta is renting out his vineyard on Airbnb . The world class midfielder posted on Twitter to spread the news . The vineyard is a snip of a price to tread paths made by one of the beautiful game 's best players at $ 125 ( £ 90 ) a night .

## Footnotes

[1] https://github.com/google-research/text-to-text-transfer-transformer