[Reviews · NeurIPS 2020]

Review 1

Summary and Contributions: This paper proposes a method for improving pretrained language models using adversarial methods. It starts off by suggesting one of the main issues is that discriminators used in previous attempts at making GANs work for NLP have been unable to learn much about specific parts of the space, because sampling generations results in outputs that span too much of the space. Instead, by (a) using a pretrained model (b) using importance sampling or Nucleus Sampling (c) clipping importance weights and (d) using memory replay, the space which the discriminator must capture shrinks significantly allowing for better training. The authors proceed to test the proposed model on unconditional and conditional language generation, showing improvement over the initial pretrained model. I have read the rebuttal.

Strengths: 1. This paper describes a novel approach to stabilizing adversarial methods for language generation.

Weaknesses: 1. This paper was not written quite clearly, causing significant confusion.

Correctness: The methods seem correct, though I had trouble understanding many key details that had to be addressed in the rebuttal.

Clarity: The paper is mostly clear, but there are some details that are passed over. For instance, in Figure 1 what generation algorithm is being used for both curves? Furthermore, is the discriminators given access to the entire prefix? If so, that means that the Teacher Forcing Generation contains only one (1) generated word, which seems like a deeply unfair comparison to something that is purely generated.

Relation to Prior Work: Yes.

Reproducibility: Yes

Additional Feedback:


Review 2

Summary and Contributions: ColdGAN uses Generative Adversarial Networks to improve natural language generation (NLG). To do so, this work carefully explains the failings of prior methods that have failed to improve as shown in Caccia et al., 2018, and focused primarily on the sampling strategy. This work identifies the Discriminator’s dependence on sampling temperature in Table 1 and then uses this insight plus improved nucleus sampling to propose training improvements. Naive application of softmax temperature, during training, would only result in rescaled parameters, so this work uses importance sampling from reinforcement learning to train the policy of interest under samples from the reduced temperature policy. This is a careful, insightful analysis which produced strong results in language modeling, question answering and abstractive summarization. I recommend acceptance at NeurIPS.

Strengths: * This research is carefully motivated, by examining specific hypotheses around Discriminator failure modes, and then it carefully employs techniques from NLP and RL to accomplish the training strategy. * Nice introduction about discriminators being able to discern human and machine generated text. Followed by a quantification of this statement with Figure 1. A discriminative model classifies accurately standard generation from teacher-forced generation. * Discriminators are specialized on a particular generative model sampling temperature. This empirically shows poor generalization of Discriminators to non-trained regimes. * Link bad reward function, per the Discriminator, to bad Generator training. Principled setup to their method versus many works that speculate first and jump to a solution. * Useful ablation study in Table 2 which details the importance of the various components of the model. Validates that standard GAN-training fails to improve and quantifies the improvements from the other aspects.

Weaknesses: * This work is motivated by exposure bias due to the mismatch of the maximum-likelihood training objective with inference sampling strategy. However, in the context of stellar results from GPT2/3, this motivation feels less strong. * Space is an issue, but I would suggest including samples in the main text. * Why did the authors not also run ColdGAN without pretraining? How does that perform relative to ScratchGAN?

Correctness: * The empirical methodology is sound, the technique is well-motivated and the empirical results hold up to scrutiny. * Correctly identifies the futility of scaling the temperature during GAN-training and employs importance sampling techniques from RL to skirt the issue. * Good use of quality-diversity measures.

Clarity: Yes, the paper is clear and logical. One nit. L57 "distribution modes" is a bit jargon-y. I'd recommend more clearly stating what you mean or defining this.

Relation to Prior Work: Yes, this training technique is novel and clearly differentiated.

Reproducibility: Yes

Additional Feedback: * Why do you use a sparse reward? * In Figure 1, is there an ability to distinguish human-generated from teacher-forced generation? * In Figure 1, why is the classification accuracy starting at 70%? ==== Post Rebuttal ==== Thank you for the additional experiments and clarification. I remain positive.


Review 3

Summary and Contributions: In this paper, a new sampling strategy is designed to improve the sampling efficiency of textual GANs. Specifically, the authors use nucleus sampling, temperature-based sampling to obtain some samples for discriminator training, and then introduce the importance weight clipping technique for generator training. ===post-rebuttal=== Thanks for your rebuttal and some of my concerns were addressed in the rebuttal. Now my main concern is the hyper-parameters/tricks used in this paper, including MLE pretraining, importance weight clipping, mixture weight. It would be more convinced if adding some detailed analysis.

Strengths: I don’t see the strong strength of this paper.

Weaknesses: There are some major flaws that prevent me from giving acceptance. (1) The novelty of this paper is rather limited. The authors have mainly changed the sampling method in the original GAN framework. I do not see the strong advantages and novelty of the new sampling method. (2) I am confused about the reason why the proposed method will benefit the discriminator training. I think the accuracy of discriminator model in Figure 1 is too high. Theoretically, once the discriminator model is well trained, the accuracy of the discriminator model should be 50%. (3) The experimental results are not convincing enough. In Table 2, the improvement gained by ColdGAN is not significant.

Correctness: No, there are several statements that are unclear.

Clarity: Yes

Relation to Prior Work: Yes

Reproducibility: Yes

Additional Feedback: Overall, the paper is well written, and model structure and training details are clearly presented. It is nice to explore new sampling method for text GANs. However, in my view, the strong advantages and novelty of the new sampling method are relatively limited. Also, I am confused about some details presented in the paper. Questions: 1. The accuracy of discriminator model in Figure 1 is too high. Theoretically, when the discriminator model is well trained, the accuracy of the discriminator model should be 50%. Do you have an additional explanation for this? 2. In Table 3, the ColdGAN and BART outperform human in "Fluency". The explanation in the paper is that humans are more inclined to use informal language. Is it possible that there is an unreasonable definition of fluency for this scenario?


Review 4

Summary and Contributions: This work considers using GANs to train text generation models to solve the exposure bias problem in autoregressive models trained with MLE. Based on the observation that the discriminator is prone to domain mismatch, this work argues that the discriminator is not providing useful reward signal at the modes of the distribution. To solve this problem, this work proposes to use importance sampling and use a temperatured distribution (or mixture with nucleus sampling) as the proposal, and weigh gradients using density ratios in the score function estimator. Combined with tricks such as pretraining with MLE, replay buffer, importance weight clipping, this work achieves better generations/diversity tradeoff than MLE on conditional generation tasks such as document summarization and question generation.

Strengths: 1. The finding that a discriminator trained on samples fails on the modes is insightful. 2. Empirically this work gets slightly better results than MLE on the tasks considered here.

Weaknesses: Major concerns: 1. While it is impressive that this work gets slightly better results than MLE, there are more hyper-parameters to tune, including mixture weight, proposal temperature, nucleus cutoff, importance weight clipping, MLE pretraining (according to appendix). I find it disappointing that so many tricks are needed. If you get rid of pretraining/initialization from T5/BART, would this method work? 2. This work requires MLE pretraining, while prior work "Training Language GANs from Scratch" does not. 3. For evaluation, since the claim of this paper is to reduce exposure bias, training a discriminator on generations from the learned model is needed to confirm if it is the case, in a way similar to Figure 1. Note that it is different from Figure 4, since during training the discriminator is co-adapting with the generator, and it might get stuck at a local optimum. 4. This work is claiming that it is the first time that language GANs outperform MLE, while prior works like seqGAN or scratchGAN all claim to be better than MLE. Is this argument based on the tradeoff between BLEU and self-BLEU from "language GANs falling short"? If so, Figure 2 is not making a fair comparison since this work uses T5/BART which is trained on external data, while previous works do not. What if you only use in-domain data? Would this still outperform MLE? Minor concerns: 5. This work only uses answer generation and summarization to evaluate the proposed method. While these are indeed conditional generation tasks, they are close to "open domain" generation rather than "close domain" generation such as machine translation. I think this work would be more convincing if it is also evaluated in machine translation which exhibits much lower uncertainties per word. 6. The discriminator accuracy of ~70% looks low to me, compared to "Real or Fake? Learning to Discriminate Machine from Human Generated Text" which achieves almost 90% accuracy. I wonder if the discriminator was not initialized with a pretrained LM, or is that because the discriminator used is too small? ===post-rebuttal=== The added scratch GAN+pretraining (and coldGAN-pretraining) experiments are fairer, but scratch GAN does not need MLE pretraining while this work does, and we know that MLE pretraining makes a big difference, so I am still not very convinced. My main concern is the existence of so many hyper-parameters/tricks: mixture weight, proposal temperature, nucleus cutoff, importance weight clipping, and MLE pretraining. I think some sensitivity analysis similar to scratch GAN's would be very helpful. In addition, rebuttal Figure 2 is weird: when generating only one word, why would cold GAN already outperform MLE by 10%? To me, this seems to imply that improvement might be due to hyper-parameter tuning.

Correctness: Yes.

Clarity: Yes.

Relation to Prior Work: Yes, but iI think "Real or Fake? Learning to Discriminate Machine from Human Generated Text" is relevant to the domain mismatch claim.

Reproducibility: Yes

Additional Feedback: 1Do you also initialize the discriminator using T5/BART? What are the discriminators' architectures in this paper?

[Author Response · NeurIPS 2020]

Thanks to all the reviewers for the insightful comments and feedback.

**- About the use of pretraining** (**R1,R2,R3,R4**) Our text GAN is the first to outperform MLE, to the best of our
knowledge (based on our results and those from the ICLR'20 paper "Language GANs falling short"). While it also
strongly outperforms previous GANs, we are the first to leverage on self-supervision pretraining (BART/T5). Hence, all
reviewers pointed out that comparison is unfair, and wonder how much comes from the pretrained language model.
First, please note that **most previous works do use MLE pretraining as us**, ScratchGAN is the exception. And **none**
**of them report results that outperform their corresponding MLE** (including ScratchGAN), as opposed to us. Also,
to further analyse the results, we complete Fig. 2 from the paper with the performance of 2 additional models (Fig. 1
below, under *rebuttal* ). We observe that i) when initialised with T5, ScratchGAN under-performs MLE, as opposed to
our proposed ColdGAN; and ii) when randomly initialised, our ColdGAN is the only GAN to outperform MLE.

Figure 1: Results on the EMNLP 2017 News dataset (for all metrics, lower is better). Scores for previous works are taken from "Training language gans from scratch". *Scratch GAN+T5*: Scratch GAN but this time initialised with pretrained weights from T5; *ColdGAN-T5*: ColdGAN, but not initialised with T5 pretrain;

Figure 2: We completed the figure with our GAN outputs in *Standard* generation mode, as requested by R4. The discriminator is less able to distinguish than when generation comes from MLE.

**- About finetuning (R1)** There is a misunderstanding here, that we feel had a strong impact in the evaluation: we
agree that without the finetuning of baselines, our experiments would be misleading. But all our experiments actually
considered models finetuned via MLE to the specific task. While it was not explicitly written in the paper, this was
somehow contained in the "MLE" name of our baselines. Please note that implementation details are given in appendix.

**- Regarding Fig. 1 (R1,R2,R3,R4)** In Fig. 1 from the paper, we used our baseline T5-small. We will make this
explicit if accepted. To answer R1, the objective of this figure was 1) to compare it with the standard generation to
highlight the exposure bias and 2) to assess the ability of the discriminator on these two modes. There is clearly an
ability to distinguish human-generated from teacher-forced generation since the accuracy is higher than 50%. To answer
R2, accuracy starts at 70 %, indicating that even with a very short prefix the discriminator is good at distinguishing real
from generated texts. To answer R3 and R4 (who have opposite feelings w.r.t. the reported results), please note that
results are consistent with previous works as written line 24, including those of the "Real or Fake?" paper suggested
by R4 (70% accuracy for T=1 corresponds to the score this paper reports for its smaller Relative prefix length). The
good accuracy of discriminators, even for short prefix, can be explained by the fact that they are trained on the output
distributions of the corresponding generators. This specialization allows them to well distinguish human from generated
texts for these specific generation distributions (but not for every modified distributions as shown in table 1 of the paper).
In Fig.2 above, we show that discriminators have eventually more difficulties to distinguish real from generated texts
with a generator resulting from our ColdGAN approach than with the MLE baseline.

**- About Improvements (R3)** The improvement for automatic metrics is not large, but i) it holds true for all of them
and across all the three tasks; ii) when changing over 10 different seed initialization, the variance of the improvement is
clearly above the last SOTA, as written line 259. Note that improvement is rarely in a large magnitude for these highly
competitive tasks; iii) the automatic metrics are arguably not perfect, which is the reason why we conducted a human
evaluation, showing that our model significantly outperforms MLE for the fluency (see table 3). For fluency (question 2
of R3), we used the following definition: sentences perceived by a human as natural and grammatically correct.

**- About Novelty (R3)** Through our preliminary analysis, we are the first to show that classic discriminators in text
GANs are not stable around the generator distribution mode, which prevents them to outperform MLE performances.
We thus propose a new training methodology, based on importance sampling to focus on areas of best interest of the
representation space, that i) is not biased and ii) succeeds at stabilizing the discriminator, hence the reward, improving
thus significantly the entire training dynamic.

[Meta-Review · NeurIPS 2020]

This paper proposes a new method to stabilizing GAN for language generation. After author rebuttal and reviewer discussion, the scores are still divergent. By the end, it received 2 reject and 2 accept recommendations. On one hand, the main criticism about this paper lies in the existence of many hyper-parameters/tricks to tune. On the other hand, the reviewers appreciate the additional clarification and experiments in the rebuttal, and think this paper provides a careful and insightful analysis on text GANs. Typically in the text GAN papers, people only perform simple experiments on unconditional text generation using LSTMs in a kind of "toy" setting. In this paper, the authors have used SOTA pre-trained language models, and worked on question generation and abstractive summarization tasks. It is non-trivial to make the proposed method improve over strong MLE pre-trained models (T5 and BART). On balance, the AC recommends accepting the paper. The authors are encouraged to consider the reviewers' comments when preparing the camera-ready version.